# Structural Basis of the Inhibition of L-Methionine γ-Lyase from *Fusobacterium nucleatum*

**DOI:** 10.3390/ijms24021651

**Published:** 2023-01-13

**Authors:** Tingting Bu, Jing Lan, Inseong Jo, Jie Zhang, Xue Bai, Shanru He, Xiaoling Jin, Lulu Wang, Yu Jin, Xiaoyu Jin, Liying Zhang, Hailong Piao, Nam-Chul Ha, Chunshan Quan, Ki Hyun Nam, Yongbin Xu

**Affiliations:** 1Department of Bioengineering, College of Life Science, Dalian Minzu University, Dalian 116600, China; 2Key Laboratory of Biotechnology and Bioresources Utilization of Ministry of Education, College of Life Science, Dalian Minzu University, Dalian 116600, China; 3Infectious Diseases Therapeutic Research Center, Korea Research Institute of Chemical Technology, Daejeon 34114, Republic of Korea; 4CAS Key Laboratory of Separation Science for Analytical Chemistry, Dalian Institute of Chemical Physics, Chinese Academy of Sciences, Dalian 116600, China; 5Department of Agricultural Biotechnology, College of Agriculture and Life Sciences, Seoul National University, Gwanak-gu, Seoul 00826, Republic of Korea; 6Department of Life Science, Pohang University of Science and Technology, Pohang 35398, Republic of Korea; 7POSTECH Biotech Center, Pohang University of Science and Technology, Pohang 35398, Republic of Korea

**Keywords:** *Fusobacterium nucleatum*, hydrogen sulfide, L-methionine γ-lyase, L-cysteine desulfidase, Fn1419, gallic acid

## Abstract

*Fusobacterium nucleatum* is a lesion-associated obligate anaerobic pathogen of destructive periodontal disease; it is also implicated in the progression and severity of colorectal cancer. Four genes (*FN0625*, *FN1055*, *FN1220*, and *FN1419*) of *F. nucleatum* are involved in producing hydrogen sulfide (H_2_S), which plays an essential role against oxidative stress. The molecular functions of Fn1419 are known, but their mechanisms remain unclear. We determined the crystal structure of Fn1419 at 2.5 Å, showing the unique conformation of the PLP-binding site when compared with L-methionine γ-lyase (MGL) proteins. Inhibitor screening for Fn1419 with L-cysteine showed that two natural compounds, gallic acid and dihydromyricetin, selectively inhibit the H_2_S production of Fn1419. The chemicals of gallic acid, dihydromyricetin, and its analogs containing trihydroxybenzene, were potentially responsible for the enzyme-inhibiting activity on Fn1419. Molecular docking and mutational analyses suggested that Gly112, Pro159, Val337, and Arg373 are involved in gallic acid binding and positioned close to the substrate and pyridoxal-5′-phosphate-binding site. Gallic acid has little effect on the other H_2_S-producing enzymes (Fn1220 and Fn1055). Overall, we proposed a molecular mechanism underlying the action of Fn1419 from *F. nucleatum* and found a new lead compound for inhibitor development.

## 1. Introduction

Anaerobic bacteria are the most common microflora of human beings and do not cause infections in the healthy host [1]. However, upon injury or trauma to the body, these bacteria cause infections that are typically suppurative. The rising antimicrobial resistance of anaerobic bacteria poses a significant threat to public health [2].

The formation of reactive oxygen species (ROS) was considered a common effector mechanism of bactericidal antibiotics [3]. Hydrogen sulfide (H_2_S) plays an essential role in combatting ROS through its antioxidant effects, which can protect the body from oxidative stress-induced diseases [4,5]. In addition, H_2_S biogenesis is a critical contributor to bacterial antibiotic tolerance and a target for multifunctional antibiotic synergists [6]. Understanding the modes of action of antibiotics and bacterial resistance mechanisms is especially important for developing alternative therapies.

The gram-negative anaerobic bacterium *Fusobacterium nucleatum* is associated with lesions in destructive periodontal disease [7]. It has also been related to the progression and severity of colorectal cancer [8]. Although it is linked to the pathogenesis of periodontal disease, it has attracted the most attention in colonic health, and the extent to which it is detrimental or beneficial to the host remains debatable. H_2_S is produced in the gut from cysteine by epithelial cells and the intestinal microbiota [9]. Biological H_2_S is produced primarily from cysteine and/or homocysteine by cystathionine β–synthase, cystathionine γ-lyase (CGL), cysteine aminotransferase, and 3-mercaptopyruvate sulfur transferase, which work either individually or in concert [10,11,12,13]. It should be noted that *F. nucleatum* produces high amounts of H_2_S [14,15].

*F. nucleatum* harbors four genes (*FN0625, FN1055, FN1220, and FN1419*) encoding enzymes involved in the production of H_2_S [16]. *FN0625* encodes aspartate aminotransferase, a pyridoxal 5′-phosphate (PLP)-dependent enzyme, which catalyzes the conversion of L-cysteine and H_2_O to H_2_S, pyruvate, and ammonium by an α, β-elimination reaction [16]. *FN1055* also encodes a PLP-dependent enzyme, a cysteine synthase, which catalyzes the conversion of L-cysteine and H_2_O to H_2_S and L-serine by a β-replacement reaction, an unusual cysteine (hydroxyl) lyase reaction. On the other hand, *FN1220* encodes a fold-type II PLP-dependent enzyme, a lanthionine synthase, which condenses two L-cysteine molecules to generate H_2_S and the uncommon amino acid L-lanthionine through a β-replacement reaction [17]. Fn1419 has been characterized as an L-methionine γ-lyase (MGL^Met^), which belongs to the γ-family of PLP-dependent enzymes and deaminates L-methionine to 2-oxobutanoate, methanethiol, and ammonia cation [16]. Fn1419 also uses L-cysteine (MGL^Cys^) to produce H_2_S with pyruvate and ammonia by α, β-elimination [16]. MGL enzymes are found in prokaryotic and eukaryotic pathogenic microorganisms but are absent from mammals [18]. Accordingly, Fn1419 could serve as a target for novel antibacterial drugs against *F. nucleatum*, i.e., for its inhibitors and/or suicidal substrates. However, the molecular mechanism underlying Fn1419 activity remains largely uncharacterized.

To better understand the molecular mechanism underlying the activity of Fn1419 to guide its inhibitor design, in the present study, we characterized the MGL^Cys^ activity of Fn1419 and determined its crystal structure at a resolution of 2.5 Å. An inhibition assay using 39 natural compounds revealed that gallic acid and dihydromyricetin inhibit the MGL^Cys^ activity of Fn1419. We also investigated the gallic acid-binding site using molecular docking and mutagenesis studies. These findings will improve our understanding of the molecular function of Fn1419 and provide insights into the design of its inhibitors.

## 2. Results

### 2.1. Characterization of Fn1419

Fn1419 was overexpressed in *Escherichia coli* and exhibited the tetrameric state on analytical size-exclusion chromatography (Figure 1a). After purification, the concentrated Fn1419 solution showed a yellowish color (Figure 1a). Since the MGL enzyme requires a PLP molecule as a cofactor [19,20], we considered that the yellow color could be the result of PLP binding to Fn1419. UV-Vis absorption of PLP solution and purified Fn1419 were analyzed to verify the binding of PLP to Fn1419. The PLP and Fn1419 solution showed absorption peaks at 420 nm and 425 nm, respectively. The wavelength of the absorption peak of Fn1419 was similar to the absorbance of PLP bound to other enzymes [21] (Figure 1b). Since PLP molecules were not added during protein expression and purification, Fn1419-bound PLP molecules were derived from *E. coli* during recombinant protein expression. Fn1419 can use L-cysteine to produce H_2_S with pyruvate and ammonia by α, β-elimination [16] (Figure 1c). To verify the MGL^Cys^ activity of recombinant Fn1419 used in this experiment, an enzyme activity assay was performed using L-cysteine as a substrate. As a result, H_2_S production was clearly observed by the reaction of Fn1419 (Figure 1d).

### 2.2. Crystal Structure of Fn1419

To better understand the molecular function of Fn1419, its crystal structure was determined. The Fn1419 crystal belongs to the trigonal space group P3_1_21, with unit-cell parameters of a = b = 95.2 Å, and c = 302.3 Å. The final model was refined to 2.5 Å resolution, with R_work_ and R_free_ of 21.6% and 27.9%, respectively (Table 1). The electron density maps of Fn1419 were well-defined for all amino acid residues, except for the disordered region between Ser39 and Gly60.

The Fn1419 monomer consists of three spatially and functionally distinct subdomains: an N-terminal domain (NTD, residues Lys5–Asp38), a PLP-binding domain (PBD, residues Asn61–Leu245), and a C-terminal domain (CTD, residues Gly246–Ile395) (Figure 2a). The NTD consists of two α-helices (α1–α2) and is involved in dimer formation. The PBD consists of a central seven-stranded β-sheet (β1–β7), which is surrounded by seven α-helices (α3–α9), and one antiparallel β-strand (β7) that forms a β-hairpin structure with β6 (Figure 2a). The CTD consists of six α-helices (α10–α15) and four β-strands (β8–β11), forms two-antiparallel β-sheets, and is involved in catalysis. Further, the B-factor of the CTD (59.50 Å^2^) is much higher than that of NTD (51.10 Å^2^) and PBD (45.79 Å^2^), indicating that the structures of NTD and PBD are more rigid than that of CTD. Of note, the B-factor of the loop between α13 and α14 of the CTD was much higher than that for any other protein segment (Appendix A). Therefore, we concluded that the CTD might be essential in recognizing and trapping the substrate.

Fn1419 monomers form a tetrameric structure with an approximate dimension of 71 × 71 × 34 Å (Figure 2b), which is consistent with size-exclusion chromatography (Figure 1a). The dimer interface of Fn1419 molecules A and B is stabilized by 25 hydrogen bond interactions (Appendix A). While the total surface area of the Fn1419 monomer is 16,149 Å^2^, the buried surface of monomers A and B is 1733 Å^2^ and 1969 Å^2^, respectively, of which 11.1% corresponds to the partner molecule. Further, eight hydrogen bond interactions are observed in the dimer interface between Fn1419 molecules A and D (Appendix A), with buried surface areas of 693 Å^2^ and 650 Å^2^ (for monomers A and D, respectively), of which 4.3% corresponds to the partner molecule. Finally, 38 hydrogen bonds and 19 salt bridge interactions are observed in the dimer interface between Fn1419 molecules A and C (Figure 2c and Appendix A), with buried surface areas of 1516 Å^2^ and 1510 Å^2^ (for monomers A and C, respectively), of which 9.4% corresponds to the partner molecule. Further, catalytic dimeric interface analysis revealed that any two monomers contact each other via large, predominantly flat regions. At the upper dimeric interface, residues are mainly distributed in six loops (loop1, loop3, loop5, loop14, loop16, and loop21) and five α-helices (α1, α4, α9, α13, and α15). The Fn1419 dimer forms a positive-charged curtain-shaped active site pocket in the PBD of two monomers, approximately 10.7 Å long and 8.4 Å wide (Figure 2c). This active site pocket is highly conserved on the conservation surface of MGL proteins, whereas other surface regions have low amino acid conservation (Figure 2d). This assembly in the active dimer might be necessary for the catalytic reaction. The Fn1419 tetrameric structure is mainly stabilized by hydrogen-bond linkages at the dimer–dimer interface and functions to maintain an active dimer.

Next, we compared the crystal structure of Fn1419 with those of PLP-dependent MGLs and CGLs. Fn1419 is highly structurally similar to representative PLP-dependent MGL proteins from *Pseudomonas putida* (PpMGL; PDB entry: 5X2W; sequence identity = 54%; r.m.s. deviation = 0.81 Å), *Citrobacter freundii* (CfMGL; 6S0C; 58%; 0.56 Å), and *Clostridium sporogenes* (CsMGL; 5DX5; 67%; 0.51 Å), and also to PLP-dependent CGL from Bacillus cereus (BcCGL; 7D7O; 49%; 0.80 Å), *Staphylococcus aureus* (SaCGL; 6KGZ; 42%; 0.90 Å), and *Pseudomonas aeruginosa* (PaCGL; 7BA4; 44%; 0.73 Å). Superimposition of the MGL and CGL proteins showed large conformation differences in PBD. In particular, the loop region between the α2- and α3-helix of the crystal structure of Fn1419, CfMGL, SaCGL, and PaCGL are disordered (Figure 3a,b), which indicates that the α2-α3 loop is highly flexible in the MGL proteins family.

### 2.3. Substrate- and Cofactor-Binding Sites of Fn1419

In MGLs, PLP- and substrate-binding sites are located between the PBD and CTD [22,23,24]. Although the PLP-bound Fn1419 solution with the yellowish color was used for the crystallization, the electron density map corresponding to the PLP molecule was not observed in the predicted PLP-binding site. This observation indicates that the crystal structure of Fn1419 is in a PLP-free state, and PLP molecules were dissociated from Fn1419 during protein crystallization.

The crystal structure of this PLP-free state of Fn1419 was compared with the crystal structure of Fn1419 complexed with LLP, a covalent complex of lysine and PLP (Fn1419-6LXU, unpublished). In Fn1419-6LXU, the PLP-binding pocket is located above the positively charged pocket and is formed by five residues (Gly86, Met87, Ser206, Thr208, and Gly213) (Figure 3a). Further, in Fn1419-6LXU, the phosphate group of LLP is coordinated by four hydrogen bonds, three of which are bifurcated. The OP1 atom of PLP has interacted with the main chains of Gly86 and Ser206 and the side chain of Thr208 by hydrogen bonds, while the OP3 atom of PLP accepts a hydrogen bond from the amide group of Met87. The lysine part of LLP has interacted with Ser206, Thr208, and Gly213 via two water molecules. In addition, a structural comparison between Fn1419 and Fn1419-6LXU reveals no significant conformational change in the residues surrounding the active site (Figure 3a).

Next, the crystal structure of Fn1419 determined in this experiment was compared with that of *Pseudomonas putida* MGL (PpMGL) with 3LM, a complex of L-methionine and PLP (PDB entry: 5X2W) [22] (Figure 3b). In the PpMGL structure, the carboxyl group of the L-methionine substrate forms three hydrogen bonds with two conserved Arg375 and Ser340 residues. The O1 atom of the L-methionine substrate accepts a hydrogen bond from the NH1 atom of Arg375. In contrast, the O2 atom interacts with the amino group of Ser340 and the guanidium group of Arg375 via two hydrogen bonds. Further, the amino group of the L-methionine substrate is stabilized by a hydrogen bond with the hydroxyl group of Tyr114. Comparative analysis of amino acid sequences and crystal structures revealed that Gly86, Met87, Tyr111, Ser206, Thr208, and Lys209 of Fn1419-7BQW are located at the same structural positions as the mentioned corresponding residues of Fn1419-6LXU and PpMGL (Figure 3a,b). The side chains of these six basic amino acids are oriented towards the pocket. In addition, amino acid sequence analysis of Fn1419 and other MGL family proteins revealed that Gly86, Tyr111, Ser206, Thr208, Lys209, Ser338, and Arg373 in Fn1419, except Met87, are highly conserved (Figure 3c and Appendix A). To verify the critical residues for MGL^Cys^ activity of Fn1419, we substituted Tyr111, Ser338, and Arg373 residues of Fn1419 with an alanine residue. Y111A, S338A, and R373A mutants completely inhibited the MGL^Cys^ enzyme activity (Figure 3d). These results demonstrated that Tyr111, S338A, and Arg373 of Fn1419 are essential for substrate binding and are involved in the catalytic reaction.

### 2.4. Screening of Leading Compounds for Fn1419 Inhibition

Inhibition of H_2_S production to control the bacterial defense against ROS is a new strategy for antibacterial drug design [6]. Further, natural products and their structural analogs have historically been essential in pharmacotherapy because they are often assumed to be better tolerated and safer to use than synthetic compound molecules. Although they have side effects, including toxicity, allergic reactions, and drug interactions, they have been reported to be used for many products [25].

To provide insights for inhibitor design, we screened 39 natural compounds, including 21 hydrophilic compounds and 18 hydrophobic compounds, as potential Fn1419 inhibitors (Figure 4a,b). Among the natural hydrophilic compounds screened, the MGL^Cys^ activity of Fn1419 was significantly inhibited by gallic acid (product reduction rate: 84.8%) and slightly reduced by potassium sorbate (5.3%), N-acetyl-cysteine (12%), and creatine monohydrate (11.2%). In contrast, the MGL^Cys^ activity of Fn1419 was enhanced by betaine-HCl (product increase rate: 24%), erythritol (16.4%), choline bitartrate (13.3%), GABA (9.9%), chondroitin sulfate (5.8%), and matrine (4.9%). L-glutamate, chlorogenic acid, citicoline sodium, cordycepin, vitamin C, niacin, inositol, xylitol, L-carnitine, phenylpiracetam, and pyridoxine-HCl did not significantly affect enzyme activity. Considering the hydrophobic compounds, dihydromyricetin (product reduced by 89.5%) significantly inhibited the MGL^Cys^ activity of Fn1419 rutin (product reduction rate: 18.2%); synephrine (32.4%), kaempferol (40.4%), and curcumin (53.9%) mildly inhibited the enzyme. In contrast, the MGL^Cys^ enzyme activity was enhanced by D-raffinose pentahydrate (product increase rate: 72.5%), L-tetrahydropal matrine (71.7%), phenibut HCl (61.8%), 5-hydroxyl tryptophan (53.0%), D-quinic acid (35.1%), hordenine HCl (32.1%), oleamide (24.6%), and noopept (21.8%). It was not significantly affected by α-lipoic acid, indole-3-carbinol, p-coumaric, acetyl L-carnitine HCl, and apigenin. Next, we checked the effects of different concentrations of gallic acid and dihydromyricetin, the two most inhibitory compounds identified, on enzyme activity (Figure 4c,d). To determine the two compounds’ inhibitory effects, we performed a titration test to the calculated 50% inhibitory concentration (IC_50_) values. As a result, the IC_50_ of gallic acid and dihydromyricetin, which had relatively high Fn1419-inhibitory activity compared to other compounds, were 121 and 23 μM, respectively, indicating significant values as a lead compound for inhibitor development.

Considering the chemical structure of gallic acid and dihydromyricetin, we proposed that the trihydroxybenzene group might be critical for this inhibitive effect. To verify this, we investigated the effect of gallic acid analogs (pyrogallic acid, methyl gallate, ethyl gallate, and gallic acid trimethyl ether) on Fn1419 (Figure 4e). The analysis confirmed that trihydroxybenzene-containing pyrogallic acid (product reduction rate: 89.8%), methyl gallate (81.5%), and ethyl gallate (81.9%) significantly decreased, whereas gallic acid trimethyl ether did not affect, the MGL^Cys^ activity of Fn1419 (Figure 4f). The calculated IC_50_ values for dihydromyricetin, gallic acid, pyrogallic acid, and ethyl gallate were 23, 121, 57, and 517 μM, respectively (Figure 4g). These observations indicate that the trihydroxybenzene group is critical for the inhibition of MGL^Cys^ activity of Fn1419, and trihydroxybenzene-containing chemicals are good candidates for Fn1419-targeted antibacterial drug design against *F. nucleatum*.

### 2.5. Identification of Fn1419-Binding Site for Trihydroxybenzene-Based Lead Compounds

To analyze the trihydroxybenzene-binding mode, we performed the docking of the gallic acid molecule in the crystal structure of Fn1419-7BQW. The analysis suggested that gallic acid forms hydrogen bonds with Gly112, Lys209, Val337, and Arg373, and that the trihydroxybenzene ring is stabilized by Pro159 via a π–π interaction (Figure 4h). The gallic acid-binding site is close to the PLP-binding site on Fn1419, but only one residue from the PLP-binding site (Lys209) is involved in the catalytic activity (Figure 4h). We also modeled docking of pyrogallic acid, methyl gallate, ethyl gallate, and gallic acid trimethyl ether with Fn1419-7BQW, which showed all these compounds were bound to the same position by the trihydroxybenzene group (Appendix A).

To confirm this molecular docking result, we generated Fn1419 variants with Gly112, Pro159, and Val337 substituted with an alanine residue and assayed their MGL^Cys^ activity with and without gallic acid (Figure 4i). Fn1419-G112A retained its enzymatic activity, indicating the inhibitory effect of gallic acid was not pronounced. By contrast, Fn1419-P159A and Fn1419-V337A lost enzyme activity without added gallic acid. These observations suggest that residues P159 and V337 of Fn1419 are related to substrate binding while Gly112 of Fn1419 is critical for gallic acid binding. Indeed, the molecular docking analysis indicated that the gallic acid-binding site of Fn1419 is close to the PLP-binding cavity (Figure 4h).

The inhibitory effect of gallic acid against other PLP-dependent H_2_S-producing enzymes of *F. nucleatum* was also tested, using Fn1220, Fn1055, and Fn0625 (Figure 5a). The activity against Fn0625 was too low (H_2_S productive rate is only 11.6% compared to Fn1419) to detect inhibition, but gallic acid did not inhibit Fn1220 and Fn1055. These results suggest that gallic acid selectively inhibits the MGL^Cys^ of Fn1419.

Next, to understand this selective inhibitory activity, the amino acid sequences and crystal structures of Fn1220 and Fn1055 were analyzed (Figure 5b and Appendix A). Sequence alignment revealed a low overall sequence similarity with Fn1419: 11% for Fn1220 and 13% for Fn1055. Of note, although the amino acid similarity in the PLP-binding domain was high, Fn1419 residues that are important for gallic acid binding (Gly112, Pro159, and Val337) were not conserved (Figure 5b). Next, the PLP- and substrate-binding pocket of Fn1419 were compared with those of Fn1220, Fn1055, and Fn0625. The PLP-binding pocket is wider than that of other proteins (Figure 5c–f), which indicates that Fn1419 is able to bind to gallic acid, unlike Fn1220, Fn1055, and Fn0625. Based on the above, we propose that the differences in the PLP-binding pocket may be the reason for the selective inhibition of Fn1419 by gallic acid. Further studies on the crystal structure of Fn1419 complexed with gallic acid are required to clarify the utility of the PLP-binding pocket as a target of drug design.

## 3. Discussion

Bacteria produce endogenous H_2_S as a defense against ROS and antibiotic-induced oxidative damage [26,27,28], and H_2_S biogenesis is a critical contributor to bacterial antibiotic tolerance and a target for versatile antibiotic potentiators [6]. In *F. nucleatum*, the MGL^Cys^ protein Fn1419 is involved in synthesizing H_2_S, together with three other enzymes (Fn1220, Fn1055, and Fn0625) [16], but there are significant differences between these enzymes [16]. The crystal structures of Fn1220 and Fn1055 have been determined by Yuichiro Kezuka et al. [29]. However, until now, the crystal structures and molecular functions of Fn1419 and Fn0625 remained largely uncharacterized. Based on these, we presented the first structural and functional characterization of Fn1419 and screened for lead compound from many natural products. As a result, we found that two natural chemicals, hydrophilic gallic acid and hydrophobic dihydromyricetin, can selectively inhibit the enzyme activity of Fn1419. In addition, we also analyzed the structural insight of Fn1419 by molecule docking to understand the interaction mechanism of gallic acid with Fn1419. The crystal structure and the discovery of lead compound in Fn1419, provide a foundation for the development of selective inhibitors for this poorly studied gene.

What is noteworthy is that we obtained the crystal structure of Fn1419. The overall structure shared structure organization with its homologues (Appendix A). A structural comparison of the apo- and PLP/substrate-bound structures of Fn1419 showed that the key amino acids at the active center for cofactor/substrate binding share substantially the same conformational constraints, which suggests our structure has the ability to bind PLP and its substrates. In addition, mutant test analyses of Fn1419 suggest that Tyr111, Ser338, and Arg373 are very important for PLP and substrate binding, which lose degradation activity toward cysteine after the mutation to Ala. We guess all that because the mutation disrupts hydrogen bonds and the active center.

Many bactericidal antibiotics kill the bacteria by stimulating the production of highly toxic hydroxyl radicals, whose production is mediated by Fe^2+^ [30]. Free cysteine could accelerate the Fe^2+^-mediated Fenton reaction by reducing Fe^3+^ to Fe^2+^ [31]. To deal with the stress, the bacteria could alter gene expression levels and metabolism, which are possibly linked to inhibition of the production of hydroxyl radicals [32], which could be an inhibitor target to screen for specific inhibitors [33]. In the study, we found that gallic acid and dihydromyricetin can significantly inhibit Fn1419 enzyme activity. Considering dihydromyricetin has low solubility and permeability, as well as poor bioavailability [34], experiments mainly focused on the gallic acid, which has many biological properties, including antimicrobial, anticancer, antioxidant, and anti-inflammatory [35]. It strongly inhibits the growth of various bacteria, including antibiotic-resistant *S. aureus*, *E. coli*, *Mannheimia haemolytica*, and *Pasteurella multocida* strains [36], by inhibiting efflux pumps and folate metabolism. Additionally, gallic acid can potentiate the efficacy of antimicrobials [37]. To verify the important role of trihydroxybenzene, we selected four gallic acid analogues, containing trihydroxybenzene or not, to detect the inhibition. This result was expected since the natural compounds containing trihydroxybenzene have an inhibitory effect on H_2_S-producing enzymes from anaerobic pathogenic bacteria, which has not been reported. Due to the low IC_50_ value of gallic acid and its analogs for Fn1419, these chemicals are not suitable for application to completely inhibit Fn1419. However, we believe that gallic acid trihydroxybenzene can be used as a starting point to design and synthesize more effective inhibitors in the future.

On the other hand, gallic acid did not inhibit the H_2_S production of Fn1220 and Fn1055. To investigate the structural characteristics of these proteins that lead to the differences in the inhibitory efficacy of gallic acid on the structures and sequences of Fn1419, we analyzed Fn1220 and Fn1055. Fn1220 and Fn1055 lack key amino acid residues involved in gallic acid binding to Fn1419 (Gly112, Pro159, and Val337 on Fn1419). Further, using molecular docking and activity assays with enzyme variants, Gly112 was identified as critical to gallic acid binding to Fn1419. However, crystal structures of the inhibitor-bound complexes should be determined to delineate the role of different amino acids of Fn1419 in its sensitivity to different inhibitors.

In summary, we found that two natural compounds, gallic acid and dihydromyricetin, selectively inhibit Fn1419. Furthermore, analysis of the inhibitory effect of gallic acid analogs on Fn1419 revealed that the trihydroxybenzene component of the molecule is potentially responsible for the observed inhibition of enzymatic activity. These observations constitute a starting point for developing more potent and selective inhibitors that could also serve as a powerful tool to clarify the biological roles of this H_2_S-producing enzyme from *F. nucleatum*.

## 4. Materials and Methods

### 4.1. Protein Preparation

The sequence encoding Fn1419 (GeneBank: AAL95612.1, residues 1–395) was obtained by PCR amplification, using the genomic DNA of *F. nucleatum* ATCC 25586 as a template. The gene was cloned into the *Nco*I and *Xho*I sites of the PET-28a (+) vector (Invitrogen, Carlsbad, CA, USA) to express a recombinant protein with his-tag at the N-terminal. The resultant plasmid was transformed into *E. coli* BL21 (DE3) cells. Transformed bacteria were grown in 1 L of LB medium containing 50 mg/mL kanamycin at 37 °C until OD_600_ of 0.6–0.8. Recombinant protein expression was induced by adding 0.5 mM isopropyl-β-D-thiogalactoside (IPTG) at 30 °C for 8 h. The cells were harvested by centrifugation and resuspended in lysis buffer (20 mM Tris-HCl, pH 8.0, 150 mM NaCl, and 2 mM β-mercaptoethanol). The cells were lysed by sonication; then, centrifugation at 13,000 rpm for 30 min at 4 °C was performed to remove the cell debris. The supernatant was preincubated with Ni^2+^-NTA affinity resin (GE Healthcare, Chicago, IL, USA) for 20 min at 4 °C and then loaded onto a column. The resin was washed with washing buffer (20 mM Tris-HCl, pH 8.0, 150 mM NaCl, 2 mM β-mercaptoethanol, and 30 mM imidazole). The recombinant protein was eluted with elution buffer (20 mM Tris-HCl, pH 8.0, 150 mM NaCl, 2 mM β-mercaptoethanol, and 250 mM imidazole). The protein was concentrated by a Vivaspin centrifugal concentrator MWCO 30 kDa (Vivaspin, Littleton, MA, USA) and loaded onto a pre-equilibrated HiLoad Superdex 200 26/60 column (GE Healthcare, Chicago, IL, USA) with pre-equilibrated buffer containing 20 mM Tris-HCl, pH 8.0, 150 mM NaCl, and 2 mM β-mercaptoethanol. The protein was concentrated to 20 mg/mL using a Vivaspin centrifugal concentrator MWCO 30 kDa (Vivaspin, Littleton, MA, USA) and stored at −80 °C. The purity of the purified protein was confirmed by 15% sodium dodecyl sulfate-polyacrylamide gel electrophoresis (SDS-PAGE) and Coomassie blue R-250 staining. The sequence encoding Fn1220 (GeneBank: AAL95416.1, residues 1–316), Fn1055 (GeneBank: AAL95251.1, residues 1–336), and Fn0625 (GeneBank: AAL94821.1, residues 1–398) was constructed, expressed, and purified using the same procedure as that used for wild-type Fn1419. Site-directed mutagenesis was performed by using two subsequent PCR reactions [38], and protein variants were expressed and purified, also using the same procedure as that used for wild-type Fn1419. The oligonucleotide sequences used for PCR reactions are listed in Appendix A.

### 4.2. Analytical Size-Exclusion Chromatography

Purified Fn1419 (500 μL; 5 mg/mL) was loaded onto a Superdex 200 10/300 GL column (GE Healthcare, USA), and the protein was eluted using a buffer containing 20 mM Tris-HCl, pH 8.0, 150 mM NaCl, and 2 mM β-mercaptoethanol, at a flow rate of 0.3 mg/mL. The molecular weights of the eluted samples were calculated based on the calibration curves by β-amylase and alcohol dehydrogenase (Gel Filtration Markers Kit for Protein Molecular Weights 12,000–200,000 Da, Sigma-Aldrich, St. Louis, MO, USA).

### 4.3. PLP Binding Analysis

Absorptions of Fn1419 (0.18 mM) in buffer containing 20 mM Tris-HCl, pH 8.0, 150 mM NaCl, and 2 mM β-mercaptoethanol, and Pyridoxal 5 -phosphate (PLP, P9255, Sigma-Aldrich) were measured using ultraviolet-visible (UV-Vis-2450) spectroscopy (Shimadzu Corporation) at 25 °C in the wavelength range of 250–600 nm.

### 4.4. Enzyme Assays

MGL^Cys^ activity of Fn1419 was measured as previously reported [16]. Fn1419 (5.75 nM) in assay buffer (40 mM potassium phosphate, pH 7.6, and 10 μM PLP) was mixed with various concentrations of L-cysteine (25, 50, 100, 200, 400, 800, or 1000 μM) and incubated for 15 min at 37 °C. H_2_S, the reaction product, was then detected based on methylene blue formation [16]. Specifically, 100 μL of solution A (the enzyme reaction solution), 20 μL of solution B (20 mM N′,N′-dimethyl-p-phenylenediamine dihydrochloride and 7.2 M HCl), and 20 μL of solution C (30 mM FeCl_3_ and 1.2 M HCl) were mixed, and the mixture incubated for 30 min at room temperature. The generated methylene blue exhibited maximum absorbance at 670 nm and was detected using a SpectraMax 190 Microplate Reader (Molecular Devices, San Jose, CA, USA) at 25 °C.

### 4.5. Crystallization, Data Collection, and Structure Determination

Fn1419 solution was concentrated to 20 mg/mL using a Vivaspin centrifugal concentrator MWCO 30 kDa (Vivaspin, Littleton, MA, USA). The initial crystallization was performed using the sitting-drop vapor-diffusion method at 22 °C with commercially available crystallization screening kits, namely, PEGRx (Hampton Research, Aliso Viejo, CA, USA). Each crystal drop was composed of 0.5 μL of protein solution (20 mM Tris-HCl, pH 8.0, 150 mM NaCl, and 2 mM β-mercaptoethanol) and 0.5 μL of reservoir solution. Finally, an Fn1419 crystal was grown in a precipitate composed of 20% (*w*/*v*) polyethylene glycol 5000, 100 mM bismuth (pH 8.5), and 200 mM sodium formate at 22 °C within 5 days. The crystal was removed from the drop and immediately flash-cooled in liquid nitrogen at −173 °C.

X-ray diffraction data of Fn1419 were collected at beamline BL19U1 at the Shanghai Synchrotron Radiation Facility (SSRF) under a nitrogen stream at −173 °C. The X-ray wavelength was 0.9792 Å. A total of 360 diffraction images were collected. The diffraction data were indexed, integrated, and scaled using the HKL2000 program [39]. Initial phases were obtained using the molecular replacement method, using the Phaser [40] with the crystal structure of CsMGL (PDB entry 5DX5) as a search model [24]. Manual model-building was performed using the COOT program [41] and refined using phenix.refinement in the PHENIX package [42]. The data collection and refinement statistics are shown in Table 1. The atomic coordinates and structure factors were deposited in the Protein Data Bank (PDB entry 7BQW). All structural figures were drawn using PyMOL [43]. The tetrameric interface of Fn1419 was calculated using a PISA server available from EBI-EMBL [44]. The surface conservation of Fn1419 was calculated using the Consurf server [45]. The sequence alignment was generated using the Clustal Omega server [46]. The model structure of Fn0625 was generated by Alphafold [47].

### 4.6. Inhibitor Screening Assay

Fn1419 (5.75 nM) in assay buffer (40 mM potassium phosphate, pH 7.6 and 10 μM PLP) was mixed with 100 μM of different classes of 21 hydrophilic compounds (betaine-HCl, erythritol, choline bitartrate, GABA, chondroitin sulfate, matrine, L-glutamine, chlorogenic acid, citicoline sodium, cordycepin, vitamin C, niacin, inositol, xylitol, L-carnitine, phenylpiracetam, pyridoxine-HCl, creatine monohydrate, N-acetyl-cysteine, potassium sorbate, and gallic acid) and 18 hydrophobic compounds (d-raffinose pentahydrate, L-tetrahydropal matrine, phenibut-HCl, 5-HTP, d-quinic acid, hordenine-HCl, oleamide, noopept, α-lipoic acid, indole-3-carbinol, p-coumaric acid, acetyl L-carnitine-HCl, apigenin, futin, synephrine, kaempferol, curcumin, and dihydromyricetin) and incubated for 20 min at 20 °C. Then, 500 μM L-cysteine was added, and the reaction was allowed to proceed for 5 min at 37 °C. The amount of produced H_2_S was detected indirectly, based on the formation of methylene blue [16]. The reaction mixture consisted of solution A (100 μL), solution B (20 μL), and solution C (20 μL) (the solution composition is described at the enzyme assays) and was incubated for 30 min at 20 °C. The amount of generated methylene blue was detected at 670 nm, using a SpectraMax 190 Microplate Reader (Molecular Devices, San Jose, CA, USA) at 25 °C. The inhibitory effect of four compounds (pyrogallic acid, methyl gallate, ethyl gallate, and gallic acid trimethyl ether) was tested using the same procedure as that used for the natural compounds. The inhibitory effects of gallic acid on Fn1220 and Fn1055 were tested using the same method as above. The concentration of both enzymes was 5.75 nM, and the concentration of gallic acid was 500 μM.

### 4.7. IC_50_ Determination

Based on the changes in the MGL^Cys^ activity of Fn1419, after treatment with dihydromyricetin, gallic acid, pyrogallic acid, methyl gallate, ethyl gallate, and gallic acid trimethyl ether for 20 min at 20 °C, the generation of methylene blue was detected at 670 nm using a SpectraMax 190 Microplate Reader (Molecular Devices, San Jose, CA, USA) at 25 °C. The concentration of the enzyme was 5.75 nM and the concentration of the six above-mentioned compounds was 500 μM. IC_50_ values were eventually calculated from non-linear regression curves fitted.

### 4.8. Molecular Docking

Molecular docking of gallic acid, pyrogallic acid, and dihydromyricetin in Fn1419 was performed using the High Ambiguity Driven Protein-Protein Docking (HADDOCK) web server [48]. The 3D structures of the three small molecules (gallic acid, pyrogallic acid, and dihydromyricetin) were obtained from the PDB. Water molecules in the crystal structure of Fn1419 (PDB: 7BQW) were removed before molecular docking. The residues (81–125, 205–225, 335–355, 372–385) were defined based on the position of the active site cavity, PLP, substrate-binding residues, and the structural characteristics of small molecules. The optimal pose was selected for further study based on the HADDOCK score.

## Figures and Tables

**Figure 1 ijms-24-01651-f001:**
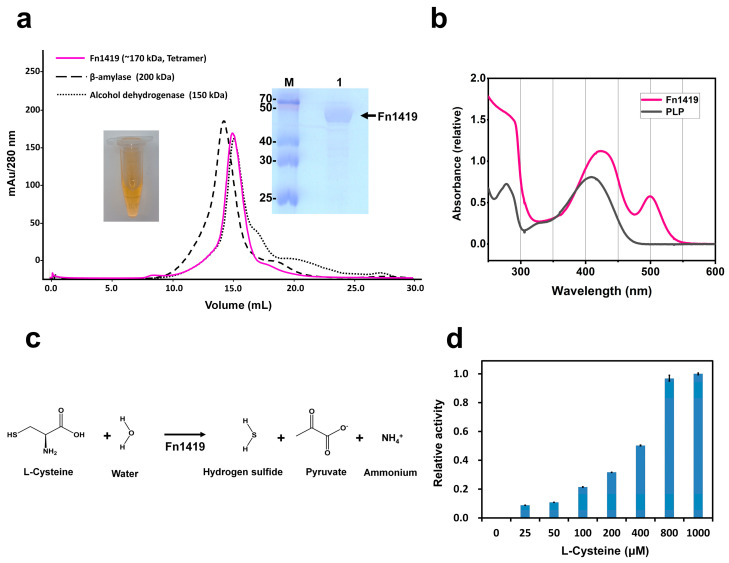
Characterization of the PLP-binding and MGL^Cys^ activity of Fn1419. (**a**) Profile of size-exclusion chromatography of Fn1419, showing the tetramer state in solution. (insert, left) The concentrated Fn1419 solution with yellow coloration. (insert, right) Purified Fn1419 on SDS-PAGE. (**b**) Spectroscopic analysis of Fn1419. The PLP and PLP-bound Fn1419 exhibited absorption peaks at 420 and 425 nm, respectively. (**c**) Reaction mechanism of *F. nucleatum* Fn1419 with L-cysteine. (**d**) The MGL^Cys^ activity of purified Fn1419 with different concentrations of L-cysteine. The error bars represent the standard deviation of three replicates.

**Figure 2 ijms-24-01651-f002:**
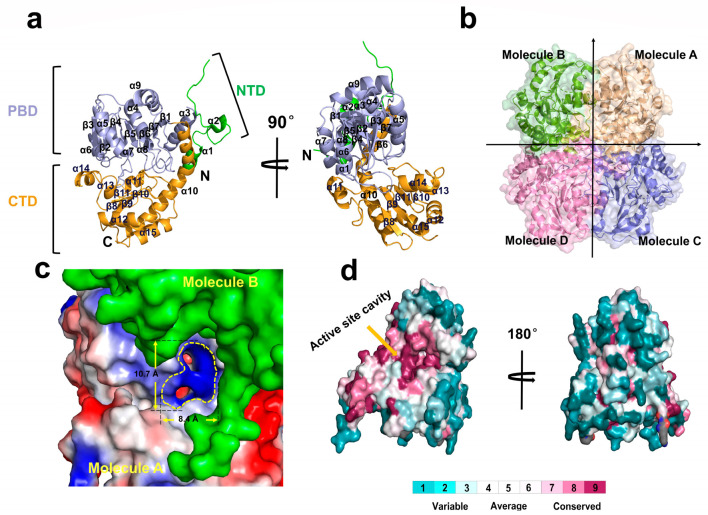
Analysis of the overall structure of Fn1419. (**a**) Monomer structure of Fn1419 consisting of the N-terminal (NTD), pyridoxal-5-phosphate–binding (PBD), and C-terminal (CTD) domains. (**b**) Tetrameric structure of Fn1419. (**c**) Electrostatistic surface of the dimer interface showing the active site. (**d**) Surface conservation of Fn1419 and MGL proteins. The active site showed high conservation, whereas the other regions were not conserved.

**Figure 3 ijms-24-01651-f003:**
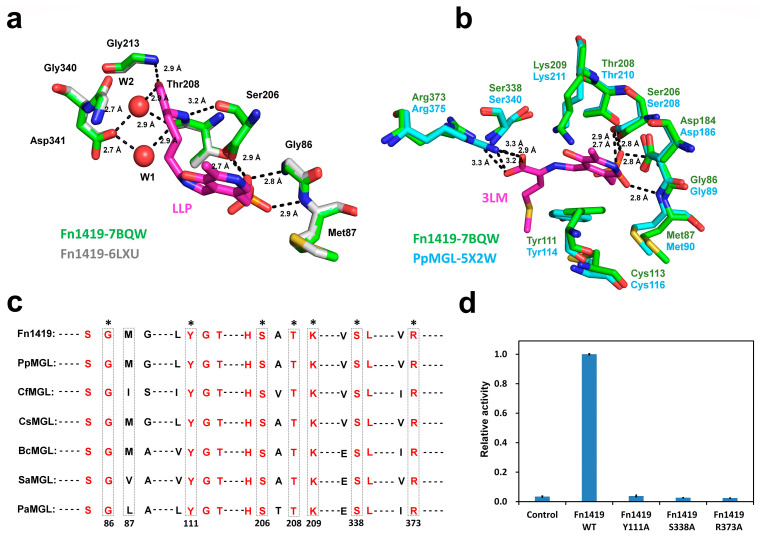
Analysis of PLP- and substrate-binding sites on Fn1419. (**a**) Superimposition of the active sites of PLP-free (PDB code: 7BQW) and PLP-bound (6LXU) states of Fn1419. (**b**) Superimposition of the active sites of Fn1419 and PpMGL-Met (5X2W). (**c**) Partial sequence alignment of MGL enzymes from seven bacteria [*Pseudomonas putida* (Uniprot accession no. P13254), *Citrobacter freundii* (Q84AR1), *Clostridium sporogenes* (A0A1J1CYV7), *Bacillus cereus* (Q818A3), *Staphylococcus aureus* (SAV0460), *and P. aeruginosa* (A0A2R3IX25)]. The PLP-binding residues are indicated by dotted boxes, and conserved residues for PLP binding are indicated by asterisks. All conserved residues shown in red. The numbering refers to Fn1419. (**d**) Measurement of the Fn1419 activity with Fn1419 mutants (Y111A, S338A, and R373A). The error bars represent the standard deviation of three replicates.

**Figure 4 ijms-24-01651-f004:**
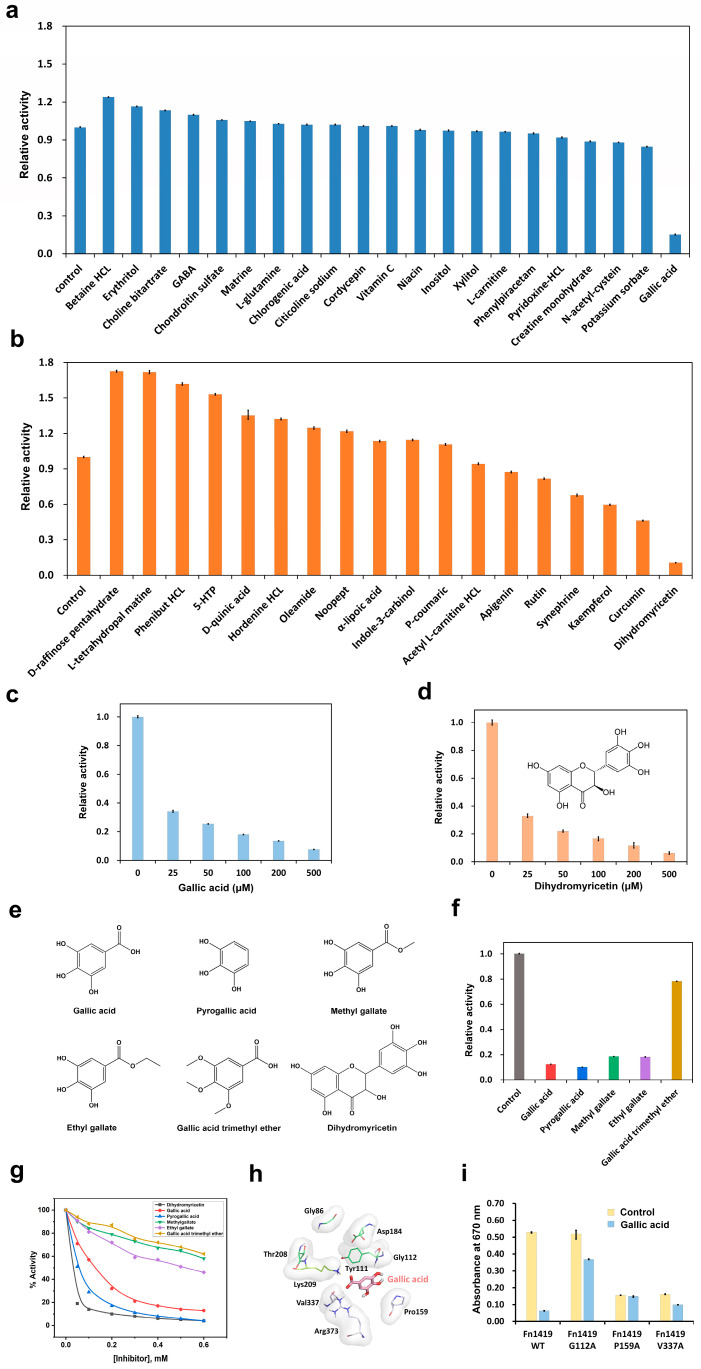
Screening for natural inhibitors of Fn1419. Inhibitor screening of (**a**) 21 hydrophilic compounds and (**b**) 18 hydrophobic compounds to Fn1419 in vitro. The error bars represent standard deviation of three replicates. Titration of (**c**) gallic acid and (**d**) dihydromyricetin to Fn1419. The error bars represent the standard deviation of three replicates. (**e**) Chemical structures of gallic acid, pyrogallic acid, methyl gallate, ethyl gallate, gallic acid trimethyl ether, and dihydromyricetin. (**f**) Fn1419 inhibition study with pyrogallic acid, methyl gallate, ethyl gallate, and gallic acid trimethyl ether. The error bars represent standard deviation of three replicates. (**g**) IC_50_ values of Fn1419 for dihydromyricetin, gallic acid, pyrogallic acid, methyl gallate, ethyl gallate, and gallic acid trimethyl ether. (**h**) Docking of gallic acid to Fn1419. Gallic acid or PLP-binding residue are indicated by gray and green sticks, respectively. (**i**) Measurement of Fn1419 activity with Fn1419 mutants (G112A, P159A, and V337A) predicted the gallic acid binding. The error bars represent standard deviation of three replicates.

**Figure 5 ijms-24-01651-f005:**
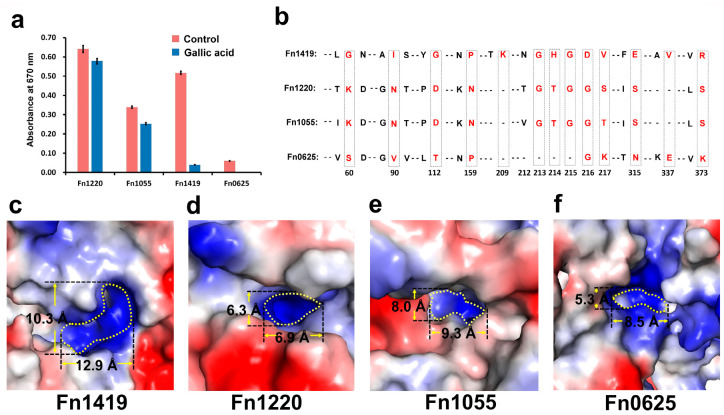
Putative gallic acid-binding sites on Fn1419. (**a**) The inhibitory effect of gallic acid on Fn1220, Fn1055, Fn1419, and Fn0625. The error bars represent the standard deviation of three replicates. (**b**) Partial sequence alignment of the active sites of Fn1220, Fn1055, Fn1419, and Fn0625. The numbering reflects Fn1419 numbering. The identified PLP- and gallic acid-binding residues are boxed and shown in red. (**c**–**f**) Electrostatistic surface of the PLP- and substrate-binding pockets of Fn1419 (PDB ID: 7BQW), Fn1220 (5XEO), Fn1055 (5B53), and Fn0625 (predicted by AlphaFold).

**Table 1 ijms-24-01651-t001:** Crystallographic data and refinement statistics.

**Data collection**	**Fn1419**
Beamline	Beamline BL19U1 at SSRF
Resolution range (Å)	42.99–2.50 (2.589–2.5)
Space group	P3_1_21
a, b, c (Å)	95.21, 95.21, 302.25
α, β, γ (o)	90.00, 90.00, 120.00
R_sym_ (%)	6.2 (30.7)
Completeness (%)	99.55 (98.89)
Multiplicity	2.1 (2.1)
Average I/σ(I)	17.27 (3.15)
**Refinement**	
R_work_/R_free_ (%)	21.60 (24.61)/27.88 (35.42)
Protein residues	395
Protein	52.73
Water	46.92
R.m.s.d. from ideal	
RMS (Bond)	0.009
RMS (Angles)	1.240
Ramachandran plot (%)	
Favored regions	94.73
Allowed regions	4.69
Disallowed regions	0.58
PDB code	7BQW

Values in parentheses are for the outermost shell.

## Data Availability

The coordinates and structure factor of Fn1419 deposited on PDB (PDB entry 7BQW). All data described in the manuscript are contained in the manuscript and the associated Appendix A files.

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
