# Peer review of "Structural Basis of the Inhibition of L-Methionine γ-Lyase from Fusobacterium nucleatum"

_ijms, 2023, doi:10.3390/ijms24021651_

Round 1
Reviewer 1 Report
Review
The authors have determined the crystal structure of Fn1419 involved in producing hydrogen sulfide (H2S) at 2.5Å, showing the unique conformation of the PLP binding site when compared with MGL proteins. And they reported that two natural compounds, gallic acid and dihydromyricetin selectively inhibit Fn1419 through inhibitor screening. Key residues for the biding with gallic acid were discovered by molecular docking and mutational analyses. The structural and analytical information found by the authors will pave the way for developing more promising and specific inhibitors for clarify the biological roles of H2S producing enzyme.
Major comments
1. The abstract is clearly written, including a basic introduction and detailed background for readers of all disciplines, while explaining the unknown physiological role of Fn1419. Structural and analytical information conducted on Fn1419 are written according to the experimental flow, and the direction to be taken in the future is also well described.
2. The main text is also well organized with a logical experimental looking from Fn1419 tetramer's overall structure, domains to active site residues. Screening for natural inhibitors of Fn1419 has significant meaning and will help in the direction of future research. The authors may provide more clarification later when adding the complex structure with promising natural inhibitors instead of docking experiments and mutational analyses.
Minor comments
1. There are a few typos in line 225 of page 7: Measuremet to Measurement, in line 109 of page 3 and line 281, 285 of page 8: errors bars to error bars, in line 395 of page 11: equlibrated to equilibrated, in line 417 of page 12: Sigmal to Sigma.
2. Subscripts should be unified in line 306, 308 of page 9: H2S to H2S
3. Superscript should be unified in line 142, 146, 147 of page 4: Å 2 to Å2
4. Scientific name of the species should be changed to italic in line 272 of page 7, in line 307 of page 9 and line 197 of page 6: F. nucleatum to F. nucleatum. and Pseudomonas putida to Pseudomonas putida
5. Please consider adding the explanation about the dotted line in Figure 1a.
6. It seems difficult to see that there is a PLP peak at 380 nm in Figure 1b. Please consider adding an auxiliary line of the peak.
7. Notation should be unified: MGLCys and MGLcys to MGLcys
Author Response
Reviewer 1
The authors have determined the crystal structure of Fn1419 involved in producing hydrogen sulfide (H2S) at 2.5Å, showing the unique conformation of the PLP binding site when compared with MGL proteins. And they reported that two natural compounds, gallic acid and dihydromyricetin selectively inhibit Fn1419 through inhibitor screening. Key residues for the biding with gallic acid were discovered by molecular docking and mutational analyses. The structural and analytical information found by the authors will pave the way for developing more promising and specific inhibitors for clarify the biological roles of H2S producing enzyme.
■ Response: Thank you for the positive feedback.
Major comments
The abstract is clearly written, including a basic introduction and detailed background for readers of all disciplines, while explaining the unknown physiological role of Fn1419. Structural and analytical information conducted on Fn1419 are written according to the experimental flow, and the direction to be taken in the future is also well described.
The main text is also well organized with a logical experimental looking from Fn1419 tetramer's overall structure, domains to active site residues. Screening for natural inhibitors of Fn1419 has significant meaning and will help in the direction of future research. The authors may provide more clarification later when adding the complex structure with promising natural inhibitors instead of docking experiments and mutational analyses.
Minor comments
There are a few typos in line 225 of page 7: Measuremet to Measurement, in line 109 of page 3 and line 281, 285 of page 8: errors bars to error bars, in line 395 of page 11: equilibrated to equilibrated, in line 417 of page 12: Sigmal to Sigma.
■ Response: Corrected.
Subscripts should be unified in line 306, 308 of page 9: H2S to H2S
■ Response: Corrected.
Superscript should be unified in line 142, 146, 147 of page 4: Å 2 to Å2
■ Response: Corrected.
Scientific name of the species should be changed to italic in line 272 of page 7, in line 307 of page 9 and line 197 of page 6: F. nucleatum to F. nucleatum. and Pseudomonas putida to Pseudomonas putida
■ Response: Corrected.
Please consider adding the explanation about the dotted line in Figure 1a.
■ Response: Corrected.
It seems difficult to see that there is a PLP peak at 380 nm in Figure 1b. Please consider adding an auxiliary line of the peak.
■ Response: We apologize for the typo. “PLP peak at 380 nm” was replaced by “PLP peak at 420 nm”. According to reviewer suggestion, we added the auxiliary line of the peak in Figure 1b as below.
Revised (Figure 1b)
Notation should be unified: MGLCys and MGLcys to MGLcys
■ Response: Corrected.

Reviewer 2 Report
Bu et. al explains the inhibition of L-met γ-lyase from F. nucleatum using mutational analysis and molecular docking studies. The study has merits. The authors are requested to respond to the comments below.
Line 27: Screening method needs to be included
Discussion:
- The authors may discuss the possible mechanism on how F. nucleatum protect oxidative stress through H2S production
- Needs to strengthen the discussion part with more information. Many results were not properly discussed.
- The possible structure activity relationship (SAR) may be discussed accordingly
Fusobacterium nucleatum and F. Nucleatum use italic throughout the manuscript
Line 80-84: the sentences are results of the study, which needs to be added to results or removed as necessary
Line 257: IC50 should be IC50
Line 257-260: The sentences needs more clarity
Supplementary Figure 5 may be placed in the main manuscript according to the corresponding result.
Author Response
Bu et. al explains the inhibition of L-met γ-lyase from F. nucleatum using mutational analysis and molecular docking studies. The study has merits. The authors are requested to respond to the comments below.
Line 27: Screening method needs to be included
■ Response: According to reviewer suggestion, we included the screen method in abstract
Original
Inhibitor screening for Fn1419 showed that two natural compounds, gallic acid, and dihydromyricetin, selectively inhibit Fn1419.
Revised (line 27-28)
Inhibitor screening for Fn1419 with L-cysteine showed that two natural compounds, gallic acid, and dihydromyricetin, selectively inhibit the H2S production of Fn1419.
Discussion:
The authors may discuss the possible mechanism on how F. nucleatum protect oxidative stress through H2S production
■ Response: According to reviewer suggestion, we added the possible mechanism as below.
Added (line 355-360)
Many bactericidal antibiotics kill the bacteria by stimulating the production of highly toxic hydroxyl radicals, whose production is mediated by Fe2+. Because free cysteine could accelerate the Fe2+-mediated Fenton reaction by reducing Fe3+ to Fe2+. To deal with the stress, the bacteria could alter gene expression levels and metabolism, which are possibly linked to inhibition of the production of hydroxyl radicals, which could be an inhibitor target to screen for specific inhibitors.
Needs to strengthen the discussion part with more information. Many results were not properly discussed.
■ Response: According to reviewer suggestion, we added the additional discussion as bellows
Added (line 346-354)
What is noteworthy is that we obtained the crystal structure of Fn1419. The overall structure shared structure organization with its homologues (Fig. S3a and S3b). A structural comparison of the apo- and PLP/substrate-bound structures of Fn1419 showed that the key amino acids at the active center for cofactor/substrate binding are share the substantially same conformational constraints, which suggest our structure has the ability to bind PLP and its substrates. In addition, mutant test analyses of Fn1419 suggest that Tyr111, Ser338 and Arg373 are very important for the PLP and substrate binding, which loses the degradation activity toward Cysteine after mutation to Ala. We guess all that be-cause of the mutation is disrupts hydrogen bonds and the active center.
The possible structure activity relationship (SAR) may be discussed accordingly
■ Response: Corrected as above (line 346-354).
Fusobacterium nucleatum and F. Nucleatum use italic throughout the manuscript
■ Response: Corrected.
Line 80-84: the sentences are results of the study, which needs to be added to results or removed as necessary
■ Response: Corrected (We removed it).
Line 257: IC50 should be IC50
■ Response: Corrected.
Line 257-260: The sentences needs more clarity
■ Response: According to reviewer suggestion, we revised the statement as below
Original
According to the result, IC50 of dihydromyricetin and gallic acid were 23 and 121 μM, respectively, indicating the two compounds have the high inhibitory activity to Fn1419 (Data see below). In conclusion, the MGLCys activity of Fn1419 was significantly reduced by gallic acid and dihydromyricetin.
Revised
As a result, the IC50 of gallic acid and dihydromyricetin, which had relatively high Fn1419 inhibitory activity compared to other compounds, were 121 and 23 μM, respectively, indicating significant values as a lead compound for inhibitor development.
Supplementary Figure 5 may be placed in the main manuscript according to the corresponding result.
■ Response: According to reviewer suggestion, we moved the Supplementary Figure 5 to Figure 4 in revised manuscript.
